# Nontuberculous Mycobacteria as Sapronoses: A Review

**DOI:** 10.3390/microorganisms10071345

**Published:** 2022-07-03

**Authors:** Ivo Pavlik, Vit Ulmann, Dana Hubelova, Ross Tim Weston

**Affiliations:** 1Faculty of Regional Development and International Studies, Mendel University in Brno, Tr. Generala Piky 7, 613 00 Brno, Czech Republic; dana.hubelova@mendelu.cz; 2Public Health Institute Ostrava, Partyzanske Nam. 7, 702 00 Ostrava, Czech Republic; vit.ulmann@zuova.cz; 3Department of Biochemistry and Genetics, La Trobe Institute for Molecular Science, La Trobe University, Science Dr, Bundoora, Melbourne, VIC 3086, Australia; r.weston@latrobe.edu.au

**Keywords:** saprophytic mycobacteria, saprozoic mycobacteria, potentially pathogenic mycobacteria, environmental saprophytic mycobacteria, geophagia, earth-eating, geochemistry, PICA practices, soil consumption, soil exposure, mud, trace elements deficiency, antidiarrheal, feeding and eating disorders

## Abstract

Mycobacteria are a unique group of microorganisms. They are characterised by exceptional adaptability and durability. They are capable of colonisation and survival even in very unfavourable conditions. In addition to the well-known obligate human pathogens, *Mycobacterium tuberculosis* and *M. leprae*, more than 200 other species have been described. Most of them form a natural part of the microflora of the external environment and thrive in aquatic and soil environments especially. For many of the mycobacterial species associated with human disease, their natural source has not yet been identified. From an ecological point of view, mycobacteria are saprophytes, and their application in human and animal diseases is opportunistic. Most cases of human disease from saprophytic mycobacteria occur in immunocompromised individuals. This adaptability and resilience to environmental pressures makes treatment of mycobacterial diseases (most often sapronoses and less often zoonoses) and permanent eradication of mycobacteria from the environment very difficult. Saprophytic mycobacterial diseases (sapronoses) are chronic and recurrent due to the fact of repeated endogenous or exogenous re-exposure. Therefore, knowledge regarding their occurrence in soil and dust would aid in the prevention of saprophytic mycobacterioses. In conjunction, their presence and ecological significance in the environment can be revealed.

## 1. Introduction

The current way of life for much of the population in both economically developed and developing countries is changing with respect to exposure to dust and soil. There is a significant trend of people making an effort to become closer to nature. In cities, the establishment of community gardens and growing plants and flowers in apartments (especially on balconies) and houses is on the rise. Furthermore, the rural surroundings of large cities are often used by the urban population for recreation and tourism. In large urban agglomerations, migration from cities to return to a life in the countryside is on the increase. This suburbanisation is also related to the development of so-called “second housing” (living in cottages, garden houses, or restored old houses). The transition to a post-industrial society, signified by the increasing number of people employed in the tertiary sector (services) with less physically demanding work than was the case previously with a decline in employment in heavy industry, is also influencing this change. These lifestyle changes are resulting in increased contact with dust and soil in nature through gardening in house gardens, in small greenhouses, in boxes on balconies, or with potting soil in households. These activities are factors for consideration related to the occurrence of threatening pathogens including nontuberculous mycobacteria (NTM) of soil-borne diseases [1,2,3,4,5,6,7,8,9,10,11,12,13].

The growing interest in soil-borne diseases is also due to the increase in the population of immunocompromised individuals as well as a general decrease in the effectiveness the immune system of many people in society today. An aging population, an increase in chronic obstructive pulmonary disease (COPD), and the more frequent occurrence of various diseases that results in a decrease in the immune competence (autoimmune diseases, HIV/ADIS pandemics, etc.) of individuals has increased the propensity of being infected by NTM [10,13,14]. Environmental related factors are also relevant, with climate change, increased environmental dust in connection with an aridisation of soils and dust erosion, and the pollution of surface and groundwater used for watering agricultural crops all playing a role in increasing the prevalence and potential to be exposed to NTM. A number of soil microbiome analyses found soil to be populated by a significant number of NTM [15,16,17]. Whilst it has long been known that aquatic environments are a source of NTM that can infect people and animals, currently there is still insufficient knowledge of the ecology and sources of human exposure to many NTM species.

The aim of this review was to summarise the importance and basic properties of naturally occurring NTM in multiple environments.

## 2. History of the Term “Sapronosis” and Relative Terms

The origin of the term sapronosis comes from the Greek word, “sapros”, meaning rotten, and “nosos”, meaning disease. The compound word “sapronosis” therefore literally means “disease caused by putrefactive bacteria”. The term “sapronosis” was first used in the 1950s [18]. In 1994, Litvin and Pushkareva used the term sapronoses in the study of possible mechanisms in the formation of epidemic variants of the causative agents present in soil or water [19]. In 1999, pathogenic bacteria *Yersinia pseudotuberculosis* and *Listeria monocytogenes* were described as pathogens of saprozoonoses, because they were capable of chemolithoautotrophic assimilation of CO_2_ and a low temperature was favourable for better absorption of CO_2_ [20]. In 2013, Iurova et al. mentioned the concept of sapronosis in an epidemiological and ecological study of *Yersinia enterocolitica* in agricultural complexes [21].

It is generally accepted that sapronosis include groups of pathogens that are primarily saprophages living in soil and other environmental compartments, which infect potential hosts by chance rather than purposefully as a specific or essential part of the organism’s life cycle [22].

### 2.1. Saprozoonosis

In 1979, the term “saprozoonosis” was used to describe the prevalence of *Aspergillus* sp. in pharyngeal swabs of different cattle and donkey handlers [23]. In 1981, this term was described as a zoonotic disease that can be transmitted by both vertebrates and invertebrates [24]. Food safety issues concerning *Yersinia pseudotuberculosis* emerged due to the ability of this sapronotic bacteria to survive and to grow in food stored at 4 °C [20]. This concept has since been adopted and used. Jaffry et al. proposed to classify zoonosis into the following groups including sapro-zoonoses [25]:Sapro-zoonoses: the pathogens are transmitted indirectly by means of a vehicle (in particular soil, water, food, or plants);Direct zoonoses: the pathogens are transmitted to humans by infected vertebrates (the causative agent is unchanged, i.e., rabies, brucellosis, or trichinosis);Cyclo-zoonoses: the pathogens require more than one vertebrate host for successful transmission and for completion of its life cycle, and there is no need for transmission of invertebrate intermediate hosts (i.e., echinococcosis or taeniasis);Meta-zoonoses: pathogen is transmitted by an invertebrate vector in which the pathogen multiplies and/or develops (i.e., leishmaniosis, human plague, and schistosomiasis);Anthroponoses: the pathogen only causes disease in humans but can be transmitted from lower vertebrates (mice, etc.) and invertebrates (fleas, etc.) to humans (i.e., human plague);Zooanthroponoses: the causative pathogens are typically found in humans but can be transmitted to animals (i.e., human tuberculosis; TB), which can be transmitted to cats, dogs, and cattle living in close contact with infected people [26,27,28].

### 2.2. Zoophilous Sapronosis

In Russia, Karimova et al. described the transmission of the causative agent of the plague, *Yersinia pestis*, as zoophilous sapronosis [29].

### 2.3. Soil-Borne Pathogens: Division According to Bultman et al.

In 2005, a proposal was published in the USA to divide soil-borne infectious agents into the following four different groups [30], which was supplemented in the following edition [31]:Permanent pathogens (present in soil permanently) are a permanent part of the soil environment, i.e., the soil microflora. Their occurrence is constant in certain areas, accompanied by their multiplication in the soil;Periodic pathogens (occurring in the soil temporarily during their development) need to spend some time outside the host as part of their life cycle;Transient pathogens (surviving but not reproducing in soil, i.e., pathogens passing through the soil) can survive in soil outside the host organism for a long time but cannot proliferate in it;Incidental pathogens (occurring in the soil by accident) occur in soil and other related environments by chance, mostly in connection with human activity. They enter the soil most commonly via human waste and poor municipal hygiene and insufficiently processed animal waste.

### 2.4. Euedaphic and Soil Transmitted Pathogens

In 2011, a significantly more simplified system for categorizing pathogens of human and animal diseases transmitted by soil was proposed, which divided pathogens into two groups [32]:
1.Euedaphic pathogens (transmitted by soil contact only);2.Soil transmitted pathogens (passive mode of soil transmission, i.e., the pathogen survives in soil while maintaining its infectivity to the host organism, but these infections are not exclusive to soil contact and other sources of infection exist).


## 3. Saprophytic and Saprozoic Mycobacteria

### 3.1. Mycobacteria as Saprophytic Bacteria

In a report published in 2000, Kazda makes extensive use of the term “saprophytic mycobacteria” for mycobacterial species that occur in the external environment [33]. In addition, mentioned is the first reported saprophytic species of mycobacteria, which was initially labelled “*Smegma Bacillus*” as early as 1885 [34]. Later, in 1899, this species was described as *M. smegmatis* [35].

### 3.2. Mycobacteria as Saprozoic Bacteria

In 2009, in a review of the ecology of mycobacteria [36], the term “saprophytic mycobacteria” was commonly used. In contrast, the term “saprozoically” was used (in only one publication) to describe the ability of *M. avium* ssp. *hominissuis* to survive in protozoan cysts (*Acanthamoeba castellanii*) in a study published by Steinert et al. [37].

Since then, mycobacteria and NTM have been described as “saprozoic” pathogens in just the one publication by Ashbolt in a review discussing the major saprozoic pathogens of engineered water systems. NTM are mentioned together with legionellae and “recognised as drinking water pathogens with the potential to cause a health burden” [38].

## 4. Mycobacteria and Sapronoses

The causative agents of human TB (i.e., *M. tuberculosis*, *M. africanum*, and *M. canettii*), Buruli ulcer (*M. ulcerans*), animal TB (*M. bovis*, *M. caprae*, *M. pinnipedii*, and *M. microti*), avian TB (*M. avium* ssp. *avium*), and human leprosy (*M. leprae*) form a group of so-called “obligatory pathogenic mycobacteria”. The group of species and subspecies of mycobacteria that cause TB in humans and animals are known as “tuberculous mycobacteria”. They are not able to grow and proliferate outside of the host organism, but they can persist in soil, water, or other components of the environment for a long time. The amount of these mycobacteria in the environment outside the host organisms decreases only very gradually [39].

In addition to these health-important species and subspecies of mycobacteria, there are many more species and subspecies that occur in the environment. From a total number of 195 recognised species and subspecies, only 8 (4.1%) belong to this group of obligatory pathogenic mycobacteria (List of Prokaryotic Names with Standing in Nomenclature; LPSN) [40]. However, the other 187 (95.9%) species and subspecies belong to the group of NTM. They mainly occur in the environment. Under certain conditions, they can cause diseases in humans and animals, called mycobacteriosis. They are therefore referred to as group of “potentially pathogenic mycobacteria” [36,41].

### 4.1. First Published Information about Mycobacteria and Sapronoses

In 2003, an extended list of pathogens that cause sapronotic diseases was published. The pathogens were divided into three basic taxonomic groups of pathogens: (1) bacteria, (2) fungi, and (3) protozoa [42]. The following bacterial pathogens were listed in the bacteria group: *Clostridium perfringens*, *Bacillus anthracis*, *Vibrio parahaemolyticus*, *Klebsiella pneumoniae*, *Pseudomonas aeruginosa*, *Burkholderia pseudomallei*, *Legionella pneumophila*, *Acinetobacter calcoaceticus*, *Corynebacterium serosis*, *Rhodococcus equi*, and *Nocardia asteroids* including *M. leprae* and *M. ulcerans*. Since this publication, there has only been limited published information about mycobacteria as causative agents of sapronoses available [43].

### 4.2. Current Research in Ecology of Mycobacteria

The ecology of TB and NTM species has received increased attention over the last two decades due primarily to the spread of human and animal TB which still has not been significantly reduced in developing countries [44,45]. Furthermore, all though incidence of mycobacterial infection is generally lower in developed countries, mycobacterial infections are not reducing either [46,47]. This is most likely due to the changing immune situation for humans in relation to mycobacterial infections. In most of these countries, widespread BCG (*Bacillus Calmette-Guérin*) vaccination in humans has ceased. In 2010, in the Czech Republic the policy of mandatory BCG vaccination was changed to be optional for at-risk individuals [48]. The increase in people with compromised immune systems due the fact of HIV/AIDS, an aging population, and lifestyle changes has likely also contributed to the persistence of mycobacterial disease [47,49,50,51].

### 4.3. Mycobacteria Present in Water, Biofilms etc.

Mycobacteria can be spread by contact with infected individuals (humans or animals) or through various materials (vehicles). These can be contaminated food, water, aerosol, dust (Figure 1), soil, etc., matrices [49]. The study of the ecology of mycobacteria has focused mainly on the mycobacteria found in water environments [5,11,52,53]. Many reviews have been published in the last decade dealing with the occurrence of mycobacteria in tap water [54,55]. Water is primarily a transporter (medium) of mycobacteria as opposed to being just a reservoir; consequently, reviews have been published on the presence of mycobacteria on many different aqueous environmental niches such as aerosols [56,57], engineered water [38], drinking water [5,54,55,58,59], water from hospitals and other healthcare settings [60,61,62,63,64,65,66], irrigation waters [67,68], aquatic biofilms [69,70,71], underground water [72,73], water sediments [38], and wastewater [52,74,75].

### 4.4. Mycobacteria Present in Soil

Soil appears to be a natural reservoir of NTM (Figure 2), as the presence of many mycobacteria species has been demonstrated in various soil samples from many places on all continents [49,76]. Their occurrence in the soil is dependent on its composition. In particular, the presence of readily available organic material can significantly affect the prevalence of NTM in soil [77]. The analysis of the species composition of NTM present in different soil types was analysed by Pavlik et al. in 2009 [78].

In the biologically active layer of the soil, NTM can be detected by microscopic examination after Ziehl–Neelsen staining as red rods. The estimated levels of mycobacteria were between 100 to 100,000/g of soil in the first thorough quantitative studies [79]. Mycobacteria can also be present in soil in connection with anthropogenic activity. Many cases of mycobacteria detection have been reported in materials that are used in gardening and agriculture such as fertiliser: compost, peat, garden substrates, manure, and slurry [49].

The distribution of certain NTM species In the soil is ubiquitous, such as *M. terrae*, *M. fortuitum*, *M. flavescens*, and members of the *M. avium* complex, which have been demonstrated to be present in the soil of various habitats of many countries on all continents [49]. Early studies indicated that representation of the different NTM species found in soil was dependent on the type of soil. For example, arable land was dominated by representatives of the *M. terrae* complex: *M. terrae*, *M. nonchromogenicum* and *M. triviale*. In contrast, *M. gordonae* and *M. scrofulaceum* [79] have been found mainly in the soil of meadows, pastures, and forests. Sandy soils have a lower incidence of NTM than other soil types [80].

The occurrence of NTM in soil is influenced by many other factors such as soil moisture, pH, the availability and composition of nutrients, the availability of trace elements, the habitat type from where the soil is located, and the surrounding vegetation [49].

### 4.5. Sources of Mycobacteria in Plants and Plants’ Tissues

Numerous reviews from the last decade on the ecology of mycobacteria often mention their occurrence in soil as well as the possibility that mycobacteria can persist in various parts of plants [5,81,82,83]. These findings have implications for the epidemiology of NTM, indicating mycobacteria could be spread via fresh plant produce.

#### 4.5.1. Internalisation of Mycobacteria in Plant Tissues

Mycobacteria can be transported from the soil through damaged tissue into plant tissue. Mycobacterial transmission in this way was first demonstrated in 1972 [84], where transport of mycobacteria into the stems of nettles through vascular tissue was observed after immersion in water inoculated with mycobacteria.

Subsequently, the internalisation of mycobacteria (i.e., *M. avium* ssp. *paratuberculosis* and *M. avium* ssp. *hominissuis*) in plant tissues (i.e., roots, stem, fruit, and leaf) of other plants was confirmed in pinto bean (*Phaseolus vulgaris*) [85], grass [86], lettuce (*Lactuca sativa*), tomato (*Solanum lycopersicum*), and radish (*Raphanus raphanistrum* ssp. *sativus*) [78]. Three members of *M. simiae* complex (i.e., *M. terramassiliense*, *M. rhizamassiliense*, and *M. numidiamassiliense*) were isolated from tomato plant roots [87]. *M. sarraceniae* and *M. helvum* were detected in pitcher plants (*Sarracenia purpurea*) found growing in sphagnum peat bogs [88].

#### 4.5.2. Surface Contamination of Plants

The surface of plants (e.g., fruits, vegetables, and cereals) can be contaminated with various types of mycobacteria present in soil, dust, and other matrices. In this way, soil mycobacteria can be transmitted to humans via edible plant produce [78].

### 4.6. Mycobacteria with Unknown Natural Source

Species of NTM, which are the cause of severe mycobacteriosis or persistent colonisation and contamination of patients and for whom their source in the environment has not yet been satisfactorily identified, remain an epidemiological phenomenon today. *M. abscessus* ssp. *a**bscessus* is one of the most important causes of persistent mycobacteriosis and due to the fact of their considerable resistance to therapeutics, is practically incurable when infections/colonisation of the lungs occurs in patients with cystic fibrosis. The horizontal transfer within habitats is still unclear for this subspecies, which is distributed worldwide, and its health impact is the same across regions. The primary well-documented source of this NTM subspecies is primarily tap water. Nevertheless, it is practically undetectable in reservoirs used for the production of drinking and service water (water reservoirs and watercourses), in soil, and in other parts of the environment [89,90].

The multiplication and increase in the concentration of *M. abscessus* ssp. *a**bscessus* apparently takes place only in artificial water mains and reservoirs (whirlpools, swimming pools, bathtubs, etc.). Evolutionary adaptation of this NTM subspecies to artificial, oligotrophic ecosystems with very low competitive pressure from other occurring microflorae can be considered. Most likely, this subspecies evolved originally from natural variants of the species *M. chelonae*, which spreads horizontally within habitats through fish, waterfowl, sediments, and other matrices from this environment. *M. abscessus* ssp. *a**bscessus* is able to colonise and contaminate aqueous solutions used in healthcare. It is characterised by high genetic and thus phenotypic plasticity and adaptability. As another primary potential source of infection, or pseudo-outbreaks in cystic fibrosis treatment facilities, can be contaminated inhalation aids, common refreshment facilities, drinking water fountains, etc. [91]. The possibility of interhuman transmission is also being considered, but this has not yet been satisfactorily confirmed [92].

Other NTM species with similar ecology are *M. xenopi* and *M. gordonae*, whose pathogenic potential is low, unlike *M. abscessus* ssp. *a**bscessus*. Nevertheless, the survival capacity of these two species is considerable in the extremely unfavourable nutritional conditions of artificial water systems. The isolation of these two species in the wild has not yet been unequivocally confirmed, but the presence in water supply systems is cosmopolitan and constant [93].

*M. chimaera* is another widely observed and studied species of NTM. The species, phenotypically almost identical to the species *M. intracellulare*, has established itself as a successful coloniser of permanently moistened plastic surfaces. Its considerable clinical significance lies in the colonisation of heater-cooler units used in cardiology. *M. chimaera* spread occurs by aerosolization and expansion of contaminated technical water as a filler in these devices into the air of operating rooms. In this way, operating rooms are contaminated. The endocarditis of patients after artificial heart valve transplantation was a clinical manifestation is an example of this. These devices were the most commonly presented as the primary source and ecological reservoir of *M. chimaer**a* [94].

However, increasingly widespread and more accurate genetic typing methods reveal a wider range of *M. chimaera*. Earlier seizures of *M. chimaer**a* in household water mains and artificial water reservoirs have been identified as *M. intracellulare* [90]. As described above, the evolution of *M. intracellulare* into a new variant (species) adapted to such a specific niche can be considered.

Most enigmatic is origin, spreading, and infection sources of species such as *M. haemophillum* and *M. genavense*. Species with the potential to infect immune-compromised adults and children and have not yet been isolated in the environment. These strains have been detected only from patients’ biological samples. It is likely that their detection and identification in more microbially loaded environmental matrices may be complicated by their specific culture requirements: increased CO_2_ tension, the presence of hemin, and a very long generation time. We can also assume their limited ability to compete in the environment with other microbes, including other types of NTM [95].

## 5. Geophagia

Geophagia is the consumption of soil and other earthy materials, both accidental and intentional by animals and humans, often resulting in an improvement in mineral nutrition [96]. It is hypothesised that geophagia is a physiological response to (a) detoxify harmful or inedible substances in foodstuffs, (b) alleviate gastrointestinal problems, most often diarrhoea, (c) supplement essential minerals due to the fact of an inadequate diet, particularly Fe and Ca, and (d) alleviate excessive acidity in the digestive tract [97].

### 5.1. Geophagia in Humans

Geophagia has been observed in many animals (e.g., cattle, monkeys, wild birds) as well as in humans, pregnant women, and children in particular [98,99,100,101,102]. Geophagia in humans does not correlate with any specific age groups, race, gender, geographical area, or period [103]. However, the practice of soil consumption is clearly more common among the poorer and more tribal orientated populations, consequently, geophagia is observed most often in tropical regions and developing countries [104].

Geophagia is a risk factor for intestinal parasitism in children in developing countries. Stool examination of 286 randomly selected children aged 1–18 in three villages in African Guinea, supplemented by information from questionnaires, confirmed the relationship between geophagia and nematode infection. A total of 53% of children were infected with at least one type of soil nematode; the results of the questionnaire confirmed the practice of geophagia in 57%, 53%, and 43% of children aged 1–5, 6–10, and 11–18, respectively. The prevalence of geophagia by age and sex of children correlated with the infectious pattern of two orally acquired and soil-borne nematodes (*Ascaris lumbricoides* and *Trichuris trichiura*). These findings demonstrate that geophagia is a significant risk factor for orally acquired nematode infections in African children [100].

A case study conducted in Tanzania, Malawi, and Kenya showed that geophagia was a common practice in women: 49% in Malawi, 43% in Kenya, and 29% in Tanzania. Whereas in men, the practice of geophagia was significantly lower: 10% in Malawi and up to 1% in Kenya and Tanzania [105]. The practice of soil consumption has been described in Nigeria and Ghana, where it is associated with efforts to alleviate the symptoms of morning sickness in pregnant women [101,106]. Pregnant women are thus exposed to potentially dangerous substances, such as bacteria, fungi, helminths and ova, radioactive materials, and toxic elemental minerals, in the soil, the occurrence and amount of which depends on the geographical location. In the Volta region of Ghana, it has been found that the cumulative effect of toxic substances, especially arsenic, lead, and nickel, can have detrimental effects on the unborn child [102].

As a result of the migration of people from areas where geophagia is a common practice, it can be culturally transferred to countries where soil consumption is more accidental (unintentional) in nature. In the United Kingdom, geophagia is associated with immigrants from South Asia [107] and West Africa [108].

In humans, NTM infections resulting from geophagia has only been speculated in the literature so far without any experimental evidence. In a localised study conducted in South Africa, Felten and Knoetze reported the presence of the species *M. fortuitum*, *M. smegmatis*, and *M. vaccae* in human sputum and in the soil of these locations; furthermore, geophagia was known to be practiced in these areas, highlighting the possibility of NTM infection via geophagia [109].

### 5.2. Geophagia in Animals

Geophagia has been observed in animals in relation to the detoxification of diet related hazardous substances or the alleviation of clinical signs of diarrhoea. An example of this was described in the monkey *Macaca mulatta*. Although the intestinal tract of these monkeys was intensively infected with endoparasites (up to 89% of animals), diarrhoea was observed in only 2% of animals, and this was due to the geophagia of clayey earth particles [110]. In another example a diet containing up to 20% kaolin significantly reduced the adsorption of toxic substances in cattle and reduced diarrhoea [111]. In wild birds, geophagia of soil with clay particles protects against intoxication from ingested poisonous seeds of various fruits [112,113].

Feed supplements (Figure 3) are given to livestock to impart important dietary properties. In particular, clay minerals, peat, and naturally sourced supplements are able to absorb toxic compounds, alleviate diarrhoea and supply missing macronutrients and micronutrients [97]. Clay minerals have advantageous properties and are commonly used for livestock nutrition [114,115,116,117]. Maintenance of an appropriately acidic pH in the intestinal tract of animals is aided by the ingestion of peat, which can be used as a feed supplement [49,118,119,120].

### 5.3. Geophagia in Animals and Nontuberculous Mycobacteria

However, there are risks associated with geophagia in livestock with respect to infection by NTM. The ingestion of peat by piglets can results in infection of their intestinal tract with NTM. After fattening, at approximately 6–8 months, tuberculous lesions can be found in the mesenteric, head, liver, and other lymph nodes of these animals (Figure 4). In some pigs, tuberculous lesions in the liver and other parenchymatous organs were demonstrated. *M. avium* ssp. *hominissuis* was the most isolated causative agent in several studies [121,122,123,124,125,126]. It is very difficult to remove NTM from naturally infected peat by any disinfection procedure currently practiced including treatment with high temperatures [127].

Supplementation with kaolin can also be risky. In one study, surface water heavily contaminated with NTM was used in the production of kaolin (float). After this kaolin was used as a feed supplement to piglets, tuberculous lesions were diagnosed in these pigs at slaughter [128].

## 6. Conditions for Survival and Multiplication of Mycobacteria in the Environment

As part of the NTM infection life cycle, they must reside outside the host organism in constant contact with the local environment for some time. Outside the host, the rate of survival of NTM is affected by various external factors. The most significant factors are environmental temperature and pH [33,41,76,78].

### 6.1. Environment Temperature

Temperature is the dominant factor that significantly affects NTM presence in the environment. The dependence of their growth rate on temperature is asymmetric. Lower temperatures only slow down the division of NTM, but do not completely stop it in most species. At extremely low temperatures, NTM can no longer divide, but can usually survive for a long period of time [129,130,131,132]. In contrast, higher temperatures increase growth rate, although temperatures higher than 75 °C results in growth retardation and cell death [133].

#### 6.1.1. Psychrophilic Mycobacterial Species

Some species of NTM have psychrophilic properties, i.e., they are able to grow and multiply at temperatures lower than 20 °C; however, these are not optimal temperatures for them. Growth at 4 °C has also been reported for *M. flavescens* [134], and slow growth at 10 °C has been reported for representatives of the *M. avium* complex and *M. scrofulaceum* [135]. Examples of temperature ranges of in vitro growth for several NTM species are shown in Table 1.

At temperatures below 20 °C, the NTM generation time is usually increased, e.g., *M. marinum* typically has a generation time of 4–6 h at 25–35 °C. The generation time extends to 6.3 h at 25 °C, to 20.2 h at 20 °C, and to more than 47 h at 15 °C. At lower temperatures, the growth of *M. marinum* was no longer detected [136].

Exposure to lower temperatures is not usually lethal to mycobacteria. Most mycobacteria can survive for a long time (months to years) at a temperature below their minimum growth temperature, although they do not multiply. Even at temperatures below freezing, the mycobacterial population freezes, but it survives. This is useful for long-term storage of mycobacteria for laboratory research [33,137,138,139].

The bovine TB causative agent *M. bovis* has been found to survive in soil longer at lower temperatures (4 °C) than at 22 °C [140]. In the USA (Glacier National Park), *M. avium* was isolated from lake water at 4.1, 7.5, and 11.5 °C and *M. gordonae* at 4.9–10.2 °C [141]. In France, in subalpine habitats, *M. chelonae*, *M. fortuitum*, *M. gordonae*, *M. malmoense*, *M. szulgai*, *M. kansasii*, and members of *M. terrae* complex were isolated at 15 °C from different matrices including soil, humus, tufa, peat, rotting wood, and sphagnum [142].

#### 6.1.2. Mesophilic Mycobacterial Species

Most species of NTM belong to the mesophilic group of microorganisms due to the optimal temperature for their multiplication being between 20 and 40 °C [40]. The temperature range for optimal growth of many mesophilic NTM has been established (Table 2).

Microorganisms, including NTM, that occur in nature usually have a wider temperature tolerance range and tend to have a lower optimum temperature. In contrast, human pathogenic microorganisms, including tuberculous mycobacteria, are adapted to exactly 37 °C (e.g., *M. tuberculosis*, *M. bovis*, and *M. caprae*). However, the causative agent of avian TB (often described as avian mycobacteriosis) in birds caused by *M. avium* ssp. *avium* prefers a higher temperature for its growth (up to 41 °C). This is most likely because birds naturally have a higher physiological temperature than mammals. The in vivo virulence of bird and human sourced NTM isolates was tested by intramuscular and intraperitoneal infection in chickens and observation of disease manifestation. Bird sourced isolates retained their virulence whilst the isolates from infected humans lost their virulence [145].

#### 6.1.3. Thermophilic Mycobacterial Species

The maximum temperature for NTM survival is typically less than 50–60 °C. However, some NTM species can be considered as thermophilic microorganisms. Tortoli [143,144] stated that the NTM species listed in Table 3 have the ability to reproduce at temperatures above 40 °C and as such can be considered thermophilic NTM.

#### 6.1.4. Extremely Thermophilic Mycobacterial Species

Extremely thermophilic bacteria are able to multiply even at temperatures exceeding the boiling point of water, i.e., up to 120 °C. The optimal temperature for their growth is generally between 90 and 110 °C, and the minimum temperature for growth is usually higher than 50 °C. According to these criteria, only few species of mycobacteria can be included in this group. *M. hassiacum* can grow in the temperature range of 30–65 °C. In the USA (Yellowstone National Park, WY), *M. avium* was isolated from thermal stream water at 40.6 °C, *M. parascrofulaceum* at 43.5 and 54.9 °C [141], and the ability to grow at temperatures as high as 55 °C has been described in *M. xenopi* [146].

Microorganisms, including NTM, that occur in nature usually have a wider temperature tolerance range and tend to have a lower optimum temperature for growth. In contrast, human pathogenic microorganisms, including TB mycobacteria (*M. tuberculosis*, *M. bovis*, *M. caprae*, and other species), are adapted to 37 °C exactly [147]. Similarly, the causative agent of avian TB in birds *M. avium* ssp. *avium* prefers a higher temperature for its growth (up to 41 °C) due to the higher physiological temperature of birds [145]. For pathogenic mycobacteria even a small deviation of optimal temperature leads to their growth inhibition or even death. The host immune system utilises this and responds to infection with increased temperature and fever to destroy these pathogens [147].

In the aquatic environment, some NTM can survive short-term temperatures as high 70 °C, *M. xenopi* is one of the most thermoresistant mycobacterial species described [9,148] and *M. hassiacum* was demonstrated to able to survive temperatures reaching 73.6 °C [133]. Heat treatment reaching pasteurisation temperature for high-temperature, short-time (HTST) at 72 °C for 15 s significantly reduces the concentration of live mycobacteria [78,149,150,151,152].

### 6.2. Environmental pH

The predominance or deficiency of hydrogen cations significantly affects the metabolism of microbes. They are required for the functionality of essential enzymes needed for the growth and division of the bacterial cell and maintenance of a neutral pH in the cytoplasm [153]. Tolerance to sub-optimal higher and lower environmental pH is a competitive advantage for a bacterial strain. Mycobacteria are generally acid tolerant. The regulation of the internal environment of the cell is ensured against external conditions primarily by the unique permeability of the mycobacterial outer membranes. The complex hydrophobic and lipoid cell wall is also an effective barrier for low molecular weight substances and ions [154].

Another mechanism that protects mycobacterial cells is the limited permeability of the plasma membrane. Transmembrane transport is very slow due to the relatively small number of membranes porins [155]. The regulatory functions of ionophores (carbonyl cyanide—m-chlorophenylhydrazone), protons (monensin and nigericin), and proton pumps (F1F0 ATPase) are some of the most important molecular mechanisms helping to stabilise the internal environment of mycobacteria including *M. tuberculosis* [156]. Specific mechanisms in individual species of mycobacteria have not yet been described; however, a wide variability in mechanisms to protect the internal parts of the cell from the outside environment can be expected within the whole genus *Mycobacterium*. Obligatory pathogenic mycobacteria (i.e., *M. tuberculosis*, *M. bovis*, and *M. leprae*) are equipped with specific mechanisms and pathways (isotuberculosinol, ureC-protein) that allow them to survive in the extreme intracellular environment of macrophages. In particular, the prevention of the acidification of the phagosome, possibly prevents the penetration of the reactive content of the lysosome into the mycobacterial cell [157].

However, even mycobacteria are not able to overcome strong and long-lasting pH mediated stress. Typically, mycobacteria can withstand short-term extremely low pH only for several hours [158]. The extent to which mycobacteria survive at different pH settings has previously been verified experimentally based on culture methods. For rapid-growing mycobacterial species (i.e., members of *M. fortuitum* group), the optimal pH range for successful growth was determined to be from 7 to 7.4, and for slow-growing mycobacteria (i.e., members of *M. avium* complex), from 5.4 to 6.8 [159].

For different mycobacterial species tolerance to pH may be decisive for the colonisation of specific substrates (different ecology of individual NTM species). Molecular mechanisms of resistance of mycobacteria to variable pH have been studied mainly in the model species *M. smegmatis* [158]. However, it is possible that other NTM species have adapted induced tolerance to conditions of extreme pH through gene transfer, not only within the genus but also among other microorganisms [160]. Mutagenesis is also one of the factors leading to the selection of strains that are more resistant to external conditions or to higher virulence. Low pH tolerant mutants are capable of long-term survival within phagocytes by regulating acidification and neutralizing their internal environment [161]. Transfer of survival-enhancing genes is also possible in both mycobacteria by phage and conjugation [162,163].

Such an example could be the widespread, ubiquitous mycobacterial species *M. avium* ssp. *hominissuis;* this NTM has been demonstrated to be highly adaptable in numerous conditions. *M. avium* ssp. *hominissuis* is most commonly found in peat bogs and sphagnum and this environment is regarded as a reservoir for this NTM [164]. The pH of this environment is typically around 4.5 and presumably this NTM is well adapted to this low pH [165].

Artificial water sources (water pipes, showerheads, boilers, equipment for storage and distribution of drinking water etc.) in which the pH is approximately 7 have also been demonstrated to be reservoirs for *M. avium* ssp. *hominissuis* [166]. This subspecies is also able to survive short-term exposure to environments with an acidic pH as low as 2 as shown by its viability after passage through the stomach of mammals [167]. Furthermore, it is also able to be adapted to various niches in the human body in which it can survive for a long time, and, subsequently, cause mycobacteriosis [161]. Similarly, representatives of the *M. fortuitum* Group are able to grow in environments with a broad pH range between 5.0 and 7.4 [159]. This trait probably contributes to the ubiquitous presence of representatives of the *M. fortuitum* Group in a wide range of natural matrices and artificial environments, including tap water, household environments, bat guano, and other matrices [167,168].

## 7. Chemical Conditions for Survival and Multiplication of Mycobacteria in the Environment

Due to the specific structure of the cell wall and plasma membrane, the permeability and uptake of biogenic elements in mycobacteria is tightly controlled. The transport of molecules into the cell is mediated by specialised transmembrane porin transporters. Their number and molecular affinity are species specific and have adapted to determine a competitive advantage in the colonisation of different substrates [155].

Low molecular weight organic compounds are preferentially absorbed by mycobacteria. Of which glycerol is most essential, for the synthesis of mycolic acids and storage triacylglycerides. A limiting nitrogen source results in the formation of triacylglyceride intraplasmic inclusions and subsequently changes in cell morphology to transition into the inactive stage to allow it to survive in conditions with poor nutrition availability [169,170,171].

The ability to form aggregations of cells into compact formations (clusters and biofilms) that provide protection from external pressures is an important factor contributing to the constant presence of mycobacteria in the environment and enabling long-term settlement of new substrates, including the tissues of the human body. In the presence of sufficient nutrients, the cells are growth-transformed and released, ensuring further horizontal spread in the ecosystem [9].

Another strategy to overcome long-term extremely unfavourable conditions may be the very recently observed phenomenon of pseudo-spore formation in *M. marinum* and *M. bovis* BCG. Due to the relatively close phylogenetic relationship with the *M. tuberculosis* complex, *M. kansasii*, and *M. ulcerans*, a similar adaptation may be possible in these species or other groups of mycobacteria [172].

In general, mycobacteria are oligotrophic, able to colonise very inanimate environments such as water pipes [173], and effectively take advantage of transitional periods of minimal nutrient levels [174]. They metabolise inorganic nitrogen molecules such as ammonia, urea, nitrates, and nitrites [175], and organic amino acids (i.e., glutamate, methionine, and asparagine). Apart from glycerol, carbon sources are simple carbohydrates, organic acids, alcohols, and aldehydes. Many mycobacterial species are enzymatically equipped to decompose higher molecular weight carbon compounds such as sterols, fatty acids, and polysaccharides (starches, cellulose, and chitin) [176]. The transition to alternative carbon sources is flexible in mycobacteria and it is associated with changes in growth, viability, and multiplication rates [177].

### 7.1. Content of Organic Carbon (OC) in the Environment

Carbon is the basic “building stone” of all organic compounds and has a crucial role in the nutrient cycle. In nature, fungi and bacteria play an essential role in the decomposition of dead plant animal tissue material, resulting in the formation of various carbon compounds. Mycobacteria are classed as chemoheterotrophic bacteria, i.e., bacteria for which organic matter is a source of carbon and energy. Decomposition of organic material by environmental NTM into consumable carbon sources contributes to the biogeochemical carbon cycle [178].

Levels of total organic carbon (TOC) are often used as an indicator of the reduction in the quality of the environment with respect to pollution by organic substances. Due to the nature of organic pollution, it is necessary to distinguish between naturally occurring inorganic carbon (IC) and the production of TOC by human activity. The most common sources of natural organic pollution (i.e., natural sources of TOC) are leachates from soils and water sediments, waste and decomposition products of plant and animals and microorganisms (e.g., bacteria and fungi). Organic matter (including TOC) of anthropogenic origin comes from sewage and industrial wastewater, agricultural waste, municipal waste landfills, etc. The determination of the amount of TOC in river and lake sediments has been studied by various groups [179,180,181,182].

The effect of TOC in the environment appears to be retrograde in relation to the persistence of mycobacteria. Their survival is more limited in a eutrophic environment, where there is competition with fast-growing representatives of other microflorae [183,184]. The minimum limit of absorbable organic carbon (AOC) for mycobacteria to grow in the environment has not yet been experimentally determined; however, it can be predicted that molecular composition rather than quantity will be crucial. Simpler organic compounds will be preferentially used by mycobacteria, and their presence is necessary for their survival.

### 7.2. Phosphorus as Important Element for Mycobacterial Growth

Another important element necessary for the growth of mycobacterial cells is phosphorus, which is essential for two basic “building blocks”, nucleotides and phospholipids. The amount of phosphorus present in artificial culture medium directly correlates with the multiplication rate and quantity of mycobacterial Colony Forming Units (CFU) of a mycobacteria culture. Mycobacteria utilise simple inorganic phosphate molecules. A lack of phosphorus triggers stress genes and metabolic pathways leading to cell aggregation and biofilm formation as similarly observed when carbon or nitrogen are limited. Concentrations of as little as 1 mL/dm^3^ of substrate are probably sufficient for the continuous growth of mycobacteria [173].

### 7.3. The Impact of Metallic Chemical Elements for Mycobacterial Growth

Metallic chemical elements are required as cofactors for many mycobacterial metabolic and structural enzymes. The most important metal in this respect is iron, the presence of which in the environment is a “basic prerequisite” for the survival of mycobacteria. The absorption of metal elements from the environment is accomplished by the production of chelating molecules (siderophores). The structurally simplest siderophores are salicylic and citric acids. More complex mycobactin (i.e., carboxymycobactin and exochelin) is produced in response to iron deficiency [185]. It plays an important role in allowing mycobacteria to remain viable within the phagocytic cells of the human cellular immune system [186].

It is becoming increasingly clear that mycobacteria are evolutionarily equipped to be extremely resistant to adverse external conditions. Their elimination in the environment is very difficult and possibly impossible.

## 8. Clinical Relevance of Mycobacteria

Microorganisms are divided into four risk groups, regarding their clinical relevance for humans and animals [187]. These are: Group 1 (microorganisms that are unlikely to cause disease in humans or animals), Group 2 (microorganisms that can cause disease in humans and/or animals, but there is usually effective prophylaxis against them and the diseases caused are treatable), Group 3 (microorganisms that cause serious human and/or animal diseases, there is usually effective prophylaxis against them and the diseases caused are treatable), and Group 4 (microorganisms that cause serious human and/or animal diseases, and there is usually no effective prophylaxis or treatment available).

At present, 195 species and subspecies of mycobacteria are validly known, which can be divided according to clinical relevance from risk groups (European Union Directive 2000/54/EC and LPSN) [40,187]. A total of 99 species are included in Risk Group 1 (50.8%; Table 4), 86 belong to Risk Group 2 (44.1%; Table 5), Risk Group 3 includes 8 species (4.1%; Table 6), and no mycobacterial species belong to Risk Group 4. For one species (*M. yongonense*) clinical relevance has not been determined.

Based on published proposals by Bultman and Jeffrey and van der Putten, bacteria can be divided into groups based on their propensity to cause sapronoses [30,32].

### 8.1. Soil-Borne Mycobacteria: Division According to Bultman et al.

Bultmann et al. divided the pathogens into four groups [30]. All mycobacterial species and subspecies could be classified into one of these four basic groups proposed.

#### 8.1.1. Permanent Pathogens (Present in Soil Permanently)

Most species and subspecies of mycobacteria, which are classified in Risk Group 1 (Table 4) and Risk Group 2 (Table 5), can be included in this group. Mycobacteriosis is rare in humans and animals by species from Risk Group 1 (Table 4) and only marginally more frequently in Risk Group 2 (Table 5).

#### 8.1.2. Periodic Pathogens (Occurring in the Soil Temporarily during Their Development)

To date, no species or subspecies of mycobacteria have been described as requiring to complete part of its life cycle in soil or another matrix to allow continued growth, pathogenicity, or virulence. Therefore, based on current knowledge, no species or subspecies of mycobacteria are included in this group.

#### 8.1.3. Transient Pathogens (Surviving but Not Reproducing in Soil; Pathogens Passing through the Soil)

This group would encompass mycobacterial species from Risk Group 2 (Table 5), particularly species and subspecies of mycobacteria that are unable to reproduce outside the host organism but can survive in various components of the environment including soil for a long time. Such species include the causative agent of avian tuberculosis/mycobacteriosis (*M. avium* ssp. *avium*, *M. avium* ssp. *silvaticum,* and *M. genavense*), paratuberculosis/Johne’s disease (*M. avium* ssp. *paratuberculosis*), and cattle farcy (*M. farcinogenes*).

#### 8.1.4. Incidental Pathogens (Occurring in the Soil by Accident)

Most of the representatives listed in Risk Group 3 (Table 6) have been identified in various components of the environment, including soil and, as such, are classified in this group. The survival of the significant pathogen *M. bovis,* in soil, water, and other matrices was demonstrated by Allen et al. [44].

### 8.2. Euedaphic and Soil Transmitted Mycobacteria: Division according to Jeffery and Van Der Putten 

The approach for the classification of sapronosis agents, proposed 6 years later by Jeffrey and van der Putten, allows for a simpler classification of species and subspecies from the genus *Mycobacterium* [32]. Many species and subspecies of mycobacteria could be categorised in two or more groups proposed by Bultamnn et al. [30], whereas the classification of mycobacteria into only two groups according to the system of Jeffrey and van der Putten allows for a simple interpretation of the occurrence of mycobacteria in the environment and possible health significance [32].

#### 8.2.1. Euedaphic Pathogens (Transmitted by Soil Only)

This group would include all species and subspecies from Risk Group 1 (Table 4), because all of them have been isolated from various components (matrices) of the environment including soil. All species from Risk Group 2 except those that are not able to reproduce outside the host organism would also be categorised as Euedaphic pathogens (Table 5).

#### 8.2.2. Soil Transmitted Pathogens (Passive Mode of Soil Transmission, i.e., That the Pathogen in the Soil Survives Longer while Maintaining Its Infectivity to the Host Organism)

Species and subspecies from Risk Group 2 that are not able to reproduce in the environment or for which this has not yet been proven would be classed as soil transmitted pathogens. Such species can be included in the second group. These are, for example, *M. avium* ssp. *avium*, *M. avium* ssp. *paratuberculosis*, *M. avium* ssp. *silvaticum*, *M. farcinogenes*, and *M. genavense* (Table 5).

## 9. Conclusions

Given the relatively new perspective on soil-borne diseases, sapronoses are “underdiagnosed” and “underreported” in both humans and animals. Sapronoses are often considered to be primarily saprophages that live in soil and other environmental compartments and decomposing organic matter. Thus, they infect potential hosts only accidentally [22]. However, particularly in developing countries, soil- and dust-borne diseases are important both medically and economically [188].

The study of the ecology of mycobacteria in this review was focused on the impact of soil and dust in their spread. This was because in most developed countries, obligatory vaccination of children with BCG vaccine against human tuberculosis has ceased. This situation has led to an increase in the importance of infections in children caused by NTM species, which occur in the soil and other components of the environment and in some circumstances are multiplying [33,48,49,78].

In 1983, the term “geonosis” was first used for the cause of listeriosis [189] and subsequently in 2009 in a short communication on the pathology of ocular zoonoses [190]. However, this concept has not been adopted in the scientific literature dealing with epidemiology, although it is well known in the Star Wars saga created by director and producer George Walton Lucas Jr. [191]. Therefore, we believe that this review will clarify this new view of nontuberculous mycobacteria as sapronoses.

## Figures and Tables

**Figure 1 microorganisms-10-01345-f001:**
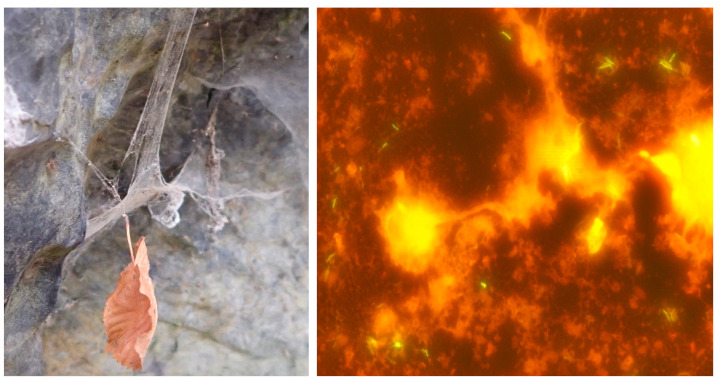
Dust in spiders’ nests (**left**) is often contaminated by mycobacteria (**right**); after the fluorescent staining of the dust sample (magnified 1000 times) yellow-coloured, rod-shaped mycobacterial cells were observed in this microscopic field of view. This sample was culture-positive for *M. triviale*.

**Figure 2 microorganisms-10-01345-f002:**
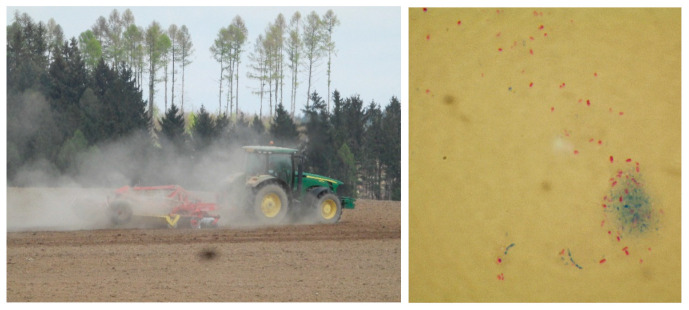
Soil (**left**) is often contaminated by mycobacteria; after Ziehl–Neelsen staining of the soil sample (magnified 1000 times) numerous acid-resistant, red-coloured polymorphic acid-fast rod-shaped mycobacterial cells were observed in this microscopic field of view (**right**). This sample was culture-positive for *M. terrae* and *M. avium* ssp. *hominissuis*.

**Figure 3 microorganisms-10-01345-f003:**
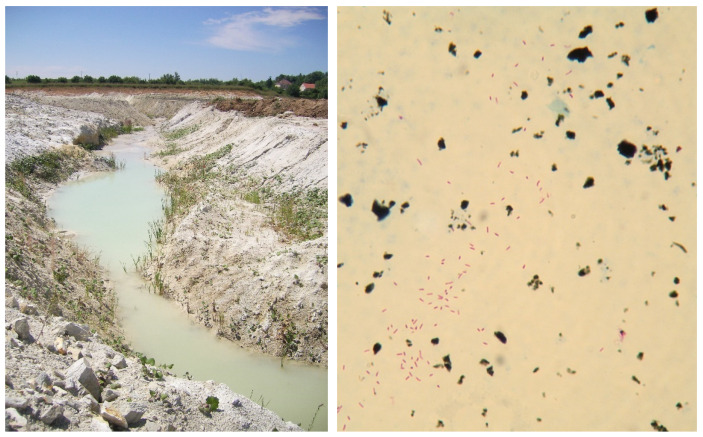
Kaolin extraction by surface mining (**left**) and direct microscopy after the Ziehl–Neelsen staining of the kaolin used as feed supplement (**right**; magnified 800 times). This sample was culture-positive for *M. avium* ssp. *hominissuis* (Photo I. Pavlik and V. Beran).

**Figure 4 microorganisms-10-01345-f004:**
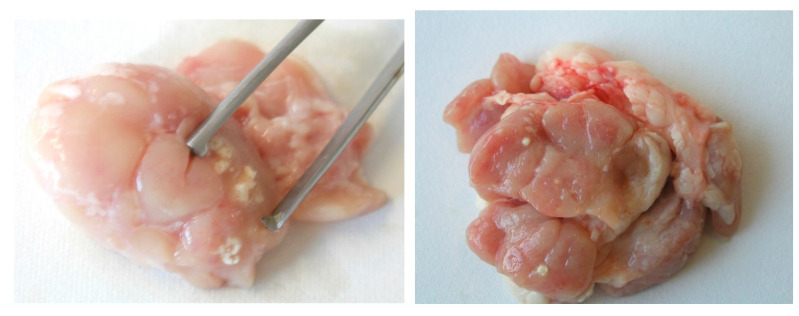
Tuberculous lesions in the mesenteric lymph (**left**) and head (**right**) nodes of domestic pig (*Sus scrofa* f. *domestica*) caused by *M. avium* ssp. *hominissuis* from peat used as feed supplements (Photo I. Pavlik).

**Table 1 microorganisms-10-01345-t001:** Psychrophilic nontuberculous mycobacteria.

Species	Temperature ^1^	Species	Temperature ^1^
** *M. arabiense* **	5–37 °C	*M. murale*	10–37 °C
*M. frederiksbergense*	15–37 °C	** *M. sediminis* **	5–37 °C
*M. hodleri*	18–37 °C		

*M*. = *Mycobacterium*. ^1^ Temperature range of in vitro growth; in plain text, the mycobacterial names are cited from Tortoli (2003) [143]; **in bold text, the mycobacterial names are cited from Tortoli (2014)** [144].

**Table 2 microorganisms-10-01345-t002:** Mesophilic nontuberculous mycobacteria.

Species	Temperature ^1^	Species	Temperature ^1^
** *M. algericum* **	25–40 °C	*M. madagascariense*	22–31 °C
** *M. alsiense* **	25–36 °C	** *M. mantenii* **	25–37 °C
*M. alvei*	25–37 °C	** *M. minnesotense* **	27–34 °C
*M. bohemicum*	25–40 °C	*M. mucogenicum*	28–37 °C
*M. brumae*	25–37 °C	** *M. noviomagense* **	37 °C
** *M. celeriflavum* **	37 °C	** *M. paraffinicum* **	22–35 °C
*M. chlorophenolicum*	18–37 °C	** *M. paragordonae* **	25–30 °C
*M. conspicuum*	22–31 °C	** *M. parakoreense* **	37 °C
*M. cookii*	22–31 °C	** *M. paraseoulense* **	25–37 °C
*M. doricum*	25–37 °C	** *M. paraterrae* **	25–37 °C
** *M. engbaekii* **	25–37 °C	*M. peregrinum*	28–35 °C
** *M. europaeum* **	30–37 °C	** *M. riyadhense* **	25–36 °C
** *M. fragae* **	30–37 °C	** *M. rufum* **	28 °C
** *M. franklinii* **	25–37 °C	** *M. salmoniphilum* **	20–30 °C
*M. fortuitum*	28–35 °C	** *M. senuense* **	25–37 °C
*M. heidelbergense*	30–37 °C	** *M. seoulense* **	25–37 °C
** *M. heraklionense* **	25–37 °C	*M. septicum*	28–35 °C
*M. hiberniae*	22–37 °C	** *M. setense* **	25–37 °C
** *M. hippocampi* **	25 °C	** *M. sherrisii* **	25–37 °C
*M. holsaticum*	20–40 °C	** *M. shigaense* **	25–37 °C
*M. immunogenum*	30–35 °C	** *M. shinjukuense* **	30–37 °C
** *M. insubricum* **	25–37 °C	*M. shottsii*	23–30 °C
** *M. iranicum* **	25–40 °C	** *M. simulans* **	37 °C
*M. interjectum*	31–37 °C	** *M. sinense* **	28–37 °C
** *M. koreense* **	25–37 °C	** *M. stomatepiae* **	25–30 °C
*M. kubicae*	28–37 °C	** *M. triplex* **	37 °C
*M. lentiflavum*	22–37 °C	*M. tusciae*	25–32 °C
** *M. liflandii* **	25–37 °C	*M. vanbaalenii*	24–37 °C
** *M. litorale* **	37 °C	** *M. yongonense* **	37 °C
** *M. llatzerense* **	22–30 °C		

*M*. = *Mycobacterium*. ^1^ Temperature range of in vitro growth; in plain text, the mycobacterial names are cited from Tortoli (2003) [143]; **in bold text, the mycobacterial names are cited from Tortoli (2014)** [144].

**Table 3 microorganisms-10-01345-t003:** Thermophilic nontuberculous mycobacteria.

Species	Temperature ^1^	Species	Temperature ^1^
** *M. arosiense* **	36–42 °C	** *M. indicus pranii* **	25–45 °C
** *M. barrassiae* **	25–42 °C	** *M. intermedium* **	25–41 °C
** *M. bouchedurhonense* **	30–45 °C	** *M. kumamotonense* **	25–42 °C
** *M. bourgelatii* **	25–42 °C	** *M. kyorinense* **	28–42 °C
*M. botniense*	36–50 °C	*M. lacus*	25–42 °C
*M. branderi*	25–45 °C	*M. mageritense*	22–45 °C
*M. celatum*	33–42 °C	** *M. marseillense* **	30–45 °C
*M. confluentis*	22–41 °C	** *M. monacense* **	25–45 °C
*M. elephantis*	25–45 °C	*M. novocastrense*	25–43 °C
*M. genavense*	25–42 °C	*M. palustre*	22–42 °C
*M. goodii*	30–45 °C	** *M. rutilum* **	28–45 °C
*M. hassiacum*	30–65 °C	** *M. timonense* **	30–45 °C
*M. heckeshornense*	30–45 °C	*M. wolinskyi*	30–45 °C

*M*. = *Mycobacterium*. ^1^ Temperature range of in vitro growth; in plain text, the mycobacterial names are cited from Tortoli (2003) [143]; **in bold text, the mycobacterial names are cited from Tortoli (2014)** [144].

**Table 4 microorganisms-10-01345-t004:** Mycobacteria in Risk Group 1 according to European Union Directive 2000/54/EC [187].

Species	Species	Species	Species	Species
*M. agri*	*M. confluentis*	*M. hiberniae*	*M. murale*	*M. rhodesiae*
*M. aichiense*	*M. cookii*	*M. hodleri*	*M. nebraskense*	*M. riyadhense*
*M. algericum*	*M. cosmeticum*	*M. holsaticum*	*M. neumannii*	*M. rufum*
*M. alvei*	*M. crocinum*	*M. insubricum*	*M. noviomagense*	*M. rutilum*
*M. aquaticum*	*M. diernhoferi*	*M. iranicum*	*M. obuense*	*M. salmoniphilum*
*M. aquiterrae*	*M. doricum*	*M. komossense*	*M. oryzae*	*M. sarraceniae*
*M. arabiense*	*M. duvalii*	*M. koreense*	*M. pallens*	*M. sediminis*
*M. arcueilense*	*M. eburneum*	*M. kumamotonense*	*M. paraffinicum*	*M. senuense*
*M. aromaticivorans*	*M. engbaekii*	*M. lacus*	*M. parafortuitum*	*M. seoulense*
*M. aurum*	*M. europaeum*	*M. lehmannii*	*M. paragordonae*	*M. shottsii*
*M. austroafricanum*	*M. fallax*	*M. litorale*	*M. parakoreense*	*M. sphagni*
*M. bohemicum*	*M. fluoranthenivorans*	*M. llatzerense*	*M. paraseoulense*	*M. stomatepiae*
*M. botniense*	*M. fragae*	*M. longobardum*	*M. paraterrae*	*M. talmoniae*
*M. bouchedurhonense*	*M. frederiksbergense*	*M. lutetiense*	*M. parmense*	*M. terrae*
*M. bourgelatii*	*M. gadium*	*M. madagascariense*	*M. phlei*	*M. thermoresistibile*
*M. brumae*	*M. gilvum*	*M. malmesburyense*	*M. poriferae*	*M. tokaiense*
*M. chitae*	*M. gordonae*	*M. minnesotense*	*M. pseudoshottsii*	*M. triviale*
*M. chlorophenolicum*	*M. grossiae*	*M. montefiorense*	*M. psychrotolerans*	*M. tusciae*
*M. chubuense*	*M. hassiacum*	*M. montmartrense*	*M. pulveris*	*M. vanbaalenii*
*M. conceptionense*	*M. helvum*	*M. moriokaense*	*M. pyrenivorans*	

*M.* = *Mycobacterium*.

**Table 5 microorganisms-10-01345-t005:** Mycobacteria in Risk Group 2 according to European Union Directive 2000/54/EC [187].

Species	Species	Species	Species	Species
*M. abscessus* (*M. abscessus* ssp. *abscessus, M. chelonae* ssp. *abscessus*)	*M. chelonae* ssp. *chelonae* (*M. chelonae*)	*M. hippocampi*	*M. mucogenicum*	*M. setense*
*M. alsense*	*M. chimaera*	*M. houstonense*	*M. neoaurum*	*M. sherrisii*
*M. arosiense*	*M. colombiense*	*M. immunogenum*	*M. neworleansense*	*M. shimoidei*
*M. arupense*	*M. conspicuum*	*M. interjectum*	*M. nonchromogenicum*	*M. shinjukuense*
*M. asiaticum*	*M. elephantis*	*M. intermedium*	*M. novocastrense*	*M. simiae*
*M. aubagnense*	*M. farcinogenes*	*M. intracellulare*	*M. palustre*	*M. smegmatis*
*M. avium* ssp. *avium* (*M. avium*)	*M. flavescens*	*M. kansasii*	*M. paraense*	*M. stephanolepidis*
*M. avium* ssp. *paratuberculosis* (*M. paratuberculosis*)	*M. florentinum*	*M. kubicae*	*M. paraintracellulare*	*M. szulgai*
*M. avium* ssp. *silvaticum*	*M. fortuitum* ssp. *acetamidolyticum*	*M. kyorinense*	*M. parascrofulaceum*	*M. timonense*
*M. bacteremicum*	*M. fortuitum* ssp. *fortuitum*	*M. lentiflavum*	*M. peregrinum*	*M. triplex*
*M. boenickei*	*M. franklinii*	*M. lepraemurium*	*M. persicum*	*M. vaccae*
*M. bolletii* (*M. abscessus* ssp. *bolletii*)	*M. gastri*	*M. mageritense*	*M. phocaicum*	*M. virginiense*
*M. branderi*	*M. genavense*	*M. malmoense*	*M. porcinum*	*M. vulneris*
*M. brisbanense*	*M. goodii*	*M. mantenii*	*M. saopaulense*	*M. wolinskyi*
*M. canariasense*	*M. haemophilum*	*M. marinum*	*M. saskatchewanense*	*M. xenopi*
*M. celatum*	*M. heckeshornense*	*M. marseillense*	*M. scrofulaceum*	
*M. celeriflavum*	*M. heidelbergense*	*M. massiliense* (*M. abscessus* ssp. *massiliense*)	*M. senegalense*	
*M. chelonae* ssp. *bovis*	*M. heraklionense*	*M. monacense*	*M. septicum*	

*M.* = *Mycobacterium*; ssp. = subspecies.

**Table 6 microorganisms-10-01345-t006:** Mycobacteria in Risk Group 3 according to European Union Directive 2000/54/EC [187].

Species	Species
*M. africanum*	*M. pinnipedii*
*M. bovis*	*M. tuberculosis* (*M. tuberculosis* ssp. *tuberculosis)*
*M. caprae* (*M. tuberculosis* ssp. *caprae*)	*M. microti*
*M. leprae*	*M. ulcerans*

*M*. = *Mycobacterium*; ssp. = subspecies.

## Data Availability

Availability of published literature and correspondence should be addressed to the corresponding author.

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
