# Peer review of "Nontuberculous Mycobacteria as Sapronoses: A Review"

_microorganisms, 2022, doi:10.3390/microorganisms10071345_

Round 1

Reviewer 1 Report

This manuscript by Pavlik et al. reviews nontuberculous mycobacteria as sapronones. The quality of the manuscript is excellent, as well as the information presented. The authors did a great job putting together this review. Especially, the background and terminology were nicely described. It definitely merits publication, almost as it is. However, it will be better if a couple of pictures are added. For example, a photo of one of those bacterias that shows its morphology, a global map of their current spread around the globe (if available), effects on humans, and animals, etc. This to make the read more fun and visual.

Author Response

We thank for the very favourable assessment of the manuscript. Four pictures were added, which demonstrate the ecology of mycobacteria in soil, dust, etc.

Reviewer 2 Report

This manuscript needs significant editing for grammar, clarity and conciseness. There's quite a bit of inconsistent use of words/phrases that can be helped by proofreading services and/or a co-author that uses english as their first language also working in this field.

The manuscript would benefit from more focus on the concepts as they relate to Mycobacteria. A great deal of the introduction and first pages are spent covering terms/definitions without any relation or reference to Mycobacteria. Bring in mycobacterial examples when possible and relate them to other bacteria or pathogens, when possible.

One notable absence from the review text is the omission of Mycobacterium abscessus. How are patients acquiring M. abscessus and inter-personal transmission of M. abscessus is one the most controversial and hottest areas of NTM research at the moment. Additionally, the potential of soil and water for M. abscessus and MAC infections are poorly discussed.

There are many recent publications on environmental NTM metagenomic and microbiome analyses in Hawaii and in national showers from the US and Europe. These were not referenced.

Author Response

We thank for all the comments that were accepted. The manuscript was significantly shortened in the introductory part. In the required sections, the manuscript was supplemented with new text and quotations. The English was re-checked by a native speaker (Ross Tim Weston), who is also a co-author of the manuscript.

Reviewer 3 Report

The manuscript is interesting, well, and clearly written.

The language doesn’t need any professional proofreading.

The title is informative, clearly stated aim tally with the content of the paper.

The abstract created an expectation that a balanced, realistic reflection on nontuberculous would be discussed in the review. The effect is achieved in the body of the manuscript.

All references were cited in the text correctly.

Apart from some punctuation mistakes, the scientific level of the review is quite high and I have no doubts to suggest the manuscript for publication. In my opinion, the presented paper might be of interest to many readerships including microbiologists, epidemiologists, and clinicians.

Author Response

We thank the third opponent for a favourable opinion. In our opinion, the additional corrections made based on the comments of the two previous opponents further improved the quality of the manuscript.